# Flow Rate Measurement of Production Profile Logging Using Thermal Method

**Yuntong Yang [1],\*, Zhaoyu Jiang [2,3], Xingbin Liu [1], Wancun Liu [4], Lianfu Han [1],\* and Lin Yang [5]**

1 School of Physics and Electronic Engineering, Northeast Petroleum University, Daqing 163318, China; dlts_liuxb@petrochina.com.cn
2 Logging and Testing Services Company, Daqing 163318, China; dlts_jiangzy@petrochina.com.cn
3 Daqing Oilfield Limited Company, Daqing 163318, China
4 Harbin Institute of Technology, School of Instrumentation Science and Engineering, Harbin 150001, China; 12b301007@hit.edu.cn
5 China National Petroleum Corporation Logging Co., LTD., Daqing 102206, China; yangl001@cnpc.com.cn
\* Correspondence: yytjzy@nepu.edu.cn (Y.Y.); lianfuhan@nepu.edu.cn (L.H.); Tel.: +86-151-6457-1168 (Y.Y.); +86-182-4962-9368 (L.H.)

**Abstract:** This paper presents a kind of thermal flow meter designed to measure downhole fluid flow at production profile logging. A computational fluid dynamics model is established to study the variation of temperature field in Downhole Thermal Flow Meter with medium and input power. The relation curve between heating power and fluid velocity and heating time is determined. According to the theoretical research, the experimental prototype of downhole thermal flow meter is designed and manufactured, and the dynamic experimental research is carried out on the multiphase flow simulation experimental device. The results show that when the power of the heating wire is constant, the temperature of the liquid around the heating wire decreases with the increase of the flow rate, and the resolution of the instrument is obvious when the flow rate is less than 20 m$^3$/d. When the flow rate is constant, the greater the power of the heating wire, the more obvious the response characteristics of the instrument. It has a good response in the whole single-phase oil and single-phase water environment. The research of theoretical and dynamic experimental shows that it is feasible to use downhole thermal flow meter to measure downhole flow. This method will provide a new idea for the measurement of flow in production profile.

**Keywords:** downhole thermal flow meter; flow rate; heating power; temperature rise; oil-water two-phase flow

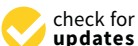



## 1. Introduction

There are many methods to measure the downhole oil-water flow at production profile well, and the accurate measurement of the flow has become an urgent problem. The fluid flowing in the well is the object of production logging flow measurement. Now, there are no moving parts, no contact and other flow velocity measurement technology has also been greatly developed. Electronic and digital flow measurement is the development trend of the field. The accurate measurement of flow is a complex problem, which is determined by the nature of flow measurement. There are many kinds of substances that flow. They can be granular rigid bodies, liquids, gases or a combination of them. The flow state of liquid flow can be laminar flow or turbulent flow. The diversity of fluid characteristic parameters determines the diversity of its measurement methods. Turbine flowmeter [1–5] is a kind of flow meter developed in the 1950s. The turbine speed is detected by non-contact magnetoelectric sensor. However, due to the limitation of the starting displacement of the turbine flowmeter, the accurate calculation of low flow production layer is affected to a certain extent. The top of the turbine is easy to wear when it rotates at high speed, causing the turbine to fall off. At the same time, it is easy to be affected by viscous resistance

and pollute. For oil and water wells with large content, the turbine turns and does not turn in actual logging; for sand return wells, the turbine is blocked by sand oil mixture. In such an environment, turbine flowmeter cannot achieve ideal effect. Conductance related flowmeter [6–10] is assumed that the fluid flow satisfies the "solidification" model. When the distance between the upstream and downstream sensors is fixed, only the transit time of the fluid flowing through the upstream and downstream sensors is needed to calculate the velocity of the fluid. However, the conductance related sensor cannot measure the flow rate of continuous phase fluid or full water. At present, this kind of flowmeter can only use the flow collection method in low production oil wells. It can't carry out the whole well measurement. Electromagnetic Flowmeter [11–14] is a non-magnetic and non-conductive pipe in the magnetic field. A pair of magnetic poles are installed outside the pipe. When the fluid with certain conductivity flows in the pipe, the magnetic line of force is cut and the induced electromotive force is generated at both ends of the conductor (flowing medium), which is derived from the electrode set on the pipe. However, the electromagnetic flowmeter can only measure the flow of conductive medium liquid, not the flow of non-conductive medium and is vulnerable to external electromagnetic interference. Thermal Mass Flowmeter [15–19] (TMF) is a kind of flow meter which measures the mass flow of fluid by using the change of temperature field produced when the fluid flows through the pipe heated by external heat source, or by using the relationship between the energy required for the fluid temperature to rise to a certain value and the mass of the fluid when heating the fluid. In industry, thermal flowmeter is generally used to measure the mass flow of gas on the ground. Yaghmourali YV et al. [20] designed a new type of suspended cantilever thermal flowmeter to measure gas flow rate. Horming Ma [21] designed a low power thermal diffusion mass air flow sensor. Guohui Lyu et al. [22] designed an electronic thermal gas flowmeter, which is used to measure the flow of gas. There are few researches on the application of thermal flowmeter to measure downhole liquid flow measurement, and there are few related literatures.

At present, turbine flowmeter, conductance correlation flowmeter and electromagnetic flowmeter are mainly used for flow measurement in oil-water well logging of production profile and stratified flow logging of horizontal well at domestic and foreign. However, due to the mechanical rotating parts of turbine flowmeter, it is easy to be stuck by downhole solid foreign matters, which leads to measurement failure and affects the success rate of logging. Conductance related flowmeter cannot measure the whole wellbore flow; electromagnetic flowmeter cannot measure the liquid with low conductivity. Therefore, it is necessary to explore a flow measurement method with no movable parts, high reliability and low flow. In this paper, a new method of downhole liquid flow measurement based on on-line heating mode is proposed, which is called Downhole Thermal Flow Meter (DTFM). This method has the advantages of high reliability, no moving parts, low flow measurement and small disturbance to fluid and can overcome the shortcomings of turbine flowmeter, and conductivity related flowmeter and electromagnetic flowmeter. In this paper, the relationship between heating power and temperature field response characteristics of DTFM is studied. The temperature field of DTFM is simulated by using computational fluid dynamics software, and the relationship curve between relative temperature and flow rate is obtained. The experimental prototype of DTFM is designed, and the experimental research is carried out on the multiphase flow simulation research device. The research provides a theoretical basis for practical production and application.

## 2. Theoretical Research

The physical basis of DTFM measurement is heat transfer. The heat dissipation in stable continuous fluid is a very complex phenomenon, because there are many physical processes at the same time. According to the theoretical of heat transfer, there are three main forms of heat transfer in the sensor element of DTFM measurement technology, including forced convection heat transfer, natural convection heat transfer and radiation heat transfer.

According to the principle of heat transfer, the electric power supplied to the temperature sensor is equal to the heat lost by the convective heat transfer of flowing fluid, the equation [23,24] can be expressed as:

$$I_w^2 R_w = hA_s(T_w - T_f) \tag{1}$$

where $I_w$ is the current passing through the probe temperature sensor, $R_w$ is the resistance of the probe temperature sensor, $h$ is the surface heat transfer coefficient, $A_s$ is the surface area of the probe, $T_w$ is the temperature of the probe temperature sensor, $T_f$ is the fluid temperature measured by the probe temperature sensor, $hA_s$ can be expressed as:

$$hA_s = A + B(q_m)^{1/2} \tag{2}$$

$A$ and $B$ are the empirical constants, $q_m$ is mass flow of fluid. According to the Equations (1) and (2), the $q_m$ can be written as:

$$q_m = \left[\frac{1}{B}\left(\frac{I_w^2 R_w}{T_w - T_f}\right) - \frac{A}{B}\right]^2 \tag{3}$$

Equation (1) can be obtained that:

$$T_w - T_f = \frac{I_w{}^2 R_w}{Bq_m^{\frac{1}{2}} + A} \tag{4}$$

It can be seen that when $T_f$ is determined, the flow rate $q_m$ is a function of $I_w$ current and temperature $T_w$. As long as either of these two parameters is fixed, the single value function relationship between the flow rate $q_m$ and the other parameter can be obtained. When the current $I_w$ is fixed, the flow rate $q_m$ can be obtained by measuring the temperature $T_w$ of the probe temperature sensor, which is a constant current mode. If the temperature $T_w$ is kept constant, that is, the temperature of the probe temperature sensor can be measured according to the relationship between the current $I_w$ through the probe temperature sensor and the flow rate $q_m$. This working mode is constant temperature mode. In this paper, the constant current working mode is adopted to study the variation law of the temperature with the flow rate.

The $q_m$ can also be expressed as:

$$q_m = \rho V S \tag{5}$$

where $\rho$ is the density of the fluid. $V$ is the velocity of the fluid, $S$ is the flow area of the fluid.

Equation (4) can be represented by:

$$q_m = f\left(\frac{I_w^2 R_w}{\Delta T}\right) \tag{6}$$

Equations (5) and (6) can be represented by:

$$V = f(\Delta T) \tag{7}$$

Through the above functional relationship, it can see that there is a certain functional relationship between the velocity and the temperature difference. As long as the temperature difference on the sensor can be measured, the fluid flow can be calculated, so as to achieve the purpose of flow measurement.

## 3. Design

### 3.1. Measurement Principle

The structure of DTFM is shown in Figure 1. The measuring device is composed of resistance heating wire, two temperature sensors and measuring circuit. The temperature sensor one is arranged under the cylindrical measuring pipe section along the fluid flow direction to measure the background temperature of the fluid. The temperature sensor two is arranged at a certain distance above the temperature sensor one, and a resistance wire is wound outside the temperature sensor two as a constant heat source. The temperature sensor two measures the degree of the resistance heating wire being cooled by the fluid under different flow rates.

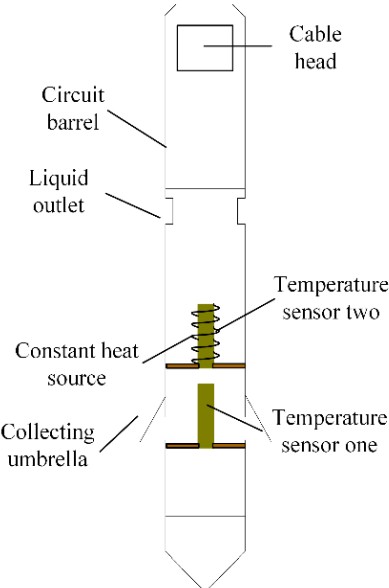

**Figure 1.** Structure diagram of DTFM.

The resistance heating wire in the measuring pipe section is used for constant current power supply. When the fluid flows through the resistance heating wire, it will take away the heat of the resistance heating wire and reduce its temperature. The temperature is measured by a highly sensitive sensor (PT1000) (Heraeus, Hanau, Germany). The larger the flow rate of liquid phase fluid, the more heat lost at the heating wire, the lower the temperature at the sensor and the smaller the resistance value of temperature sensor two. According to this principle, the flow rate of fluid can be detected by measuring the change of resistance value of two temperature sensors.

Figure 2 shows the DTFM physical diagram designed for feasibility experiment. The device is composed of circuit barrel, liquid outlet, probe short circuit and collecting umbrella from left to right. The hardware part of the experiment was placed in the circuit barrel. The liquid outlet can flow the collected liquid out of the measuring device and into the wellbore through this hole. A probe is placed in the probe short circuit, and the probe is composed of a speed sensor and a temperature sensor. The function of the collecting umbrella is to close the annular space between the measuring device and the wellbore, so that all the fluid flows through the measuring channel of the prototype, and at the same time to improve the velocity of fluid. The DTFM is used for flow measurement by flow collection. The outer diameter of the prototype is 28 mm and the inner diameter is 20 mm.

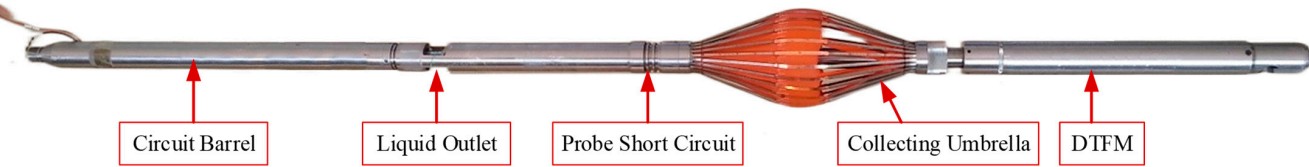

**Figure 2.** DTFM physical diagram.

In the dynamic experimental, the measuring instrument is put into the wellbore of the multiphase flow experimental device with cable. After the current collector is opened, all the fluid flows through the velocity measuring probe and the fluid flows out through the liquid outlet. The standard flow and water content are adjusted on the platform of flow control system, and the output voltage of the circuit is recorded.

*3.2. Design of Probe*

The probe of DTFM is composed of velocity sensor and temperature sensor. The velocity sensor measures the flow rate of fluid, and the temperature sensor measures the background temperature of the environment in oil-water wells to correct the measured flow rate.

Figure 3 shows the speed sensor, which is composed of heating wire covered with thermistor. Figure 4 shows the actual figure of the velocity probe. Using DC power analyzer Agilent n6705b (Keysight Technologies, Santa Rosa, CA, USA) to provide constant current for resistance heating wire. The temperature sensor used in the DTFM is composed of armored temperature sensitive platinum resistor (PT1000). Fluke pm6306 LCR measuring instrument (Fluke industrial, Almelo, the Netherlands) is used to measure the resistance of platinum resistance in real time. The resistance wire is heated under a certain power and the power is kept constant. The temperature sensor detects the temperature of the liquid phase flow at the heating wire.

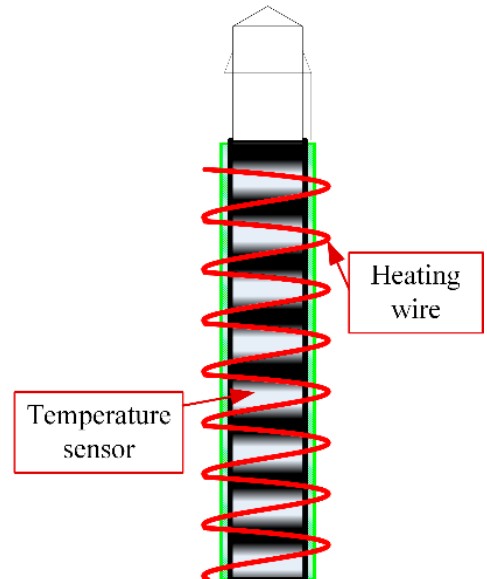

**Figure 3.** Structure diagram of temperature sensor at DTFM.

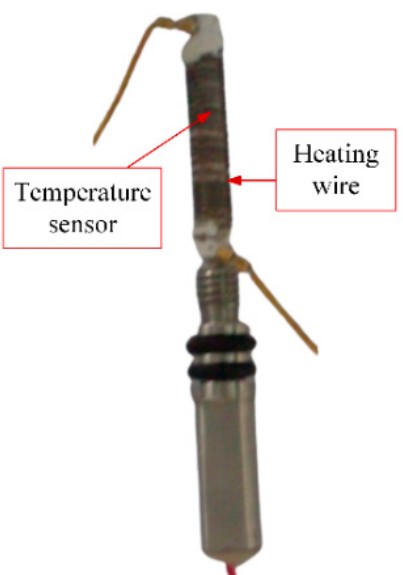

**Figure 4.** Physical diagram of temperature sensor at DTFM.

*3.3. Temperature Sensor*

The DTMF probe is shown in Figure 5. The temperature sensor one is wrapped with a heating wire, and ceramic isolation is used between the heating wire and the thermistor to prevent the heating wire directly heating the thermistor. The temperature sensor two is used to measurement the environment temperature. The temperature sensor one and tow are platinum resistance PT1000.

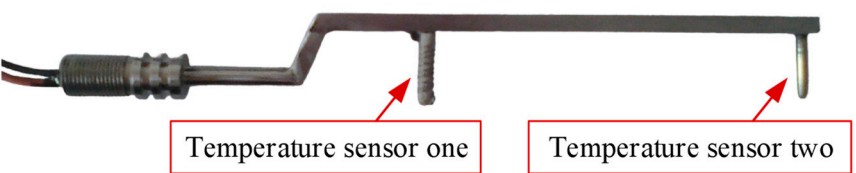

**Figure 5.** Physical image of DTMF probe.

Platinum is a naturally occurring white precious metal with high density and ductility. It is the best material for making thermal resistance, and it has strong corrosion resistance even at high temperature. The resistance body of platinum resistance is made of pin wire, which is wound on mica, quartz or ceramic support according to certain rules and covered with porcelain tube insulation. Platinum thermistor has many advantages, such as stable performance, good repeatability and high measurement accuracy. Moreover, there is a very approximate linear relationship between resistance and temperature. The disadvantage is that the temperature coefficient of resistance is small and the price is relatively expensive (uranium is a precious metal). The measuring range of standard resistance thermometer made of platinum resistance is generally $-200\,°C \sim 650\,°C$.

The relationship between the resistance of uranium and temperature is as follows:

$$R(t) = \begin{cases} R_0(1 + At + Bt^2) & 0°C \leq t \leq 850°C \\ R_0\left[1 + At + Bt^2 + Ct^3(t - 100)\right] & -200°C \leq t \leq 0°C \end{cases} \tag{8}$$

where $A$, $B$, $C$ is the constant, that is $A = 3.8623 \times 10^{-3}\,\Omega/°C$, $B = -6.5315 \times 10^{-7}\,\Omega/°C$, $C = -4.22 \times 10^{-12}\,\Omega/°C$, $R(t)$ is the resistance value of the resistance at $t$ °C, $R_0$ is the resistance value at 0 °C. In this paper the $R_0$ is 1000 $\Omega$.

In order to improve the measurement effect at low flow rate, the collecting umbrella device is used to increase the rate of the fluid flow. The fluid passes through a circular

measuring pipe section, and a thermal probe is fixed in the flow direction of the fluid. In order to ensure full contact with the fluid and prevent the heat flow body produced by heating wire from affecting the background temperature of the temperature sensor measuring environment, the temperature sensor is located upstream of the flow channel and the heating wire and measuring sensor are located downstream of the flow channel. In the experiment, the heating wire is heated by the stabilized voltage supply, when the temperature increased the resistance value of the two temperature probes changes. Thus, a corresponding voltage change is obtained at the end of the measuring circuit. According to the corresponding relationship between voltage and flow, the specific flow can be obtained.

*3.4. Measuring Circuit*

The schematic diagram of DTFM temperature measurement circuit is shown in the Figure 6. In the process of measurement, the constant current power supply is applied to the resistance heating wire. The heat dissipation of resistance heating wire in water is related to the flow rate, and the change of heat dissipation leads to the temperature change of resistance heating wire. When the temperature changes, the resistance of the speed sensor changes and the flow signal is converted into an electrical signal. Two temperature sensors are packaged in the probe, one is used to measure the background temperature of the environment and the other one is used to measure the heat dissipation of the heating wire. When the flow changes, the resistance of the temperature sensor changes accordingly. The signal acquisition module of the measurement circuit collects the resistance changes of the two sensors by building a bridge circuit at the front end, and then amplifies the collected two signals by a differential amplifier to get the voltage signal of the temperature difference changing with the flow.

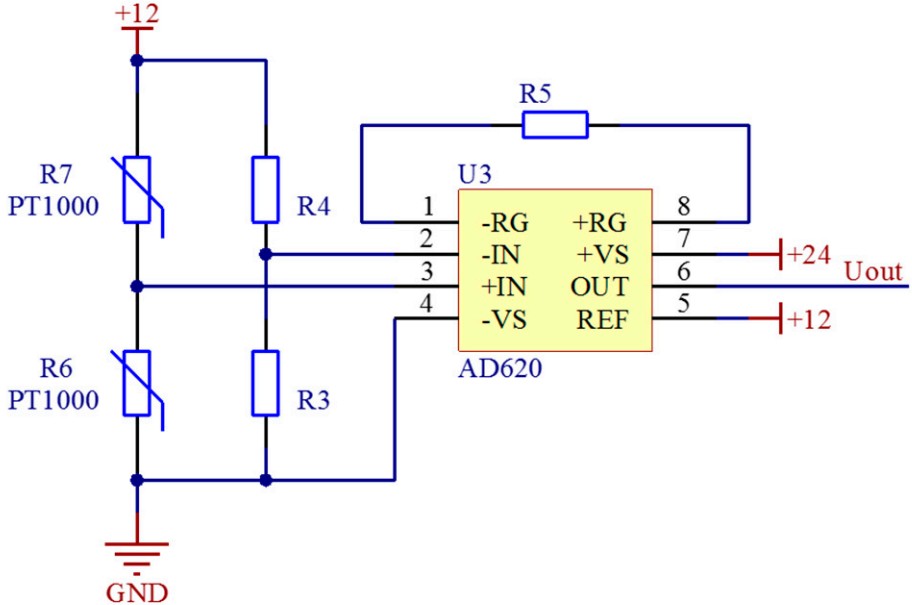

**Figure 6.** Schematic diagram of temperature measurement circuit.

The chip of signal acquisition module is AD620 (ADI, Norwood, UK). The resistance of AD620 is 11 K$\Omega$. According to the formula $G = 49.4k\Omega/R_G + 1$, the magnification is about 5.5 times. The front end of AD620 is connected with a bridge composed of two temperature sensitive resistors and two high-precision fixed resistors with equal resistance. Two temperature measuring thermistors are symmetrically distributed on both sides and form a Wheatstone bridge with two external resistors. When there is fluid flowing through, the temperature near the heating resistance wire will be higher than the temperature behind, which will change the resistance of the two thermistors, destroy the balance of the

measuring bridge and produce the output voltage ($u_{out}$) corresponding to the resistance change. The relationship of $u_{out}$ between *R6* and *R7* is

$$u_{out} = 5.5(\frac{R6}{R6 + R7} - \frac{R3}{R3 + R4}u_s)$$

(9)

where *R6* and *R7* is the PT1000, and $u_s$ = 12 V.

## 4. Simulation Study on Temperature Rise of Heating Wire Changing with Flow Rate

According to the actual physical model, the model of DTFM is established by using GAMBIT and FLUENT software (ANSYS, Inc., Canonsburg, PA, USA). The model is symmetrical, a two-dimensional model is established to simulate the distribution law between the velocity and temperature.

### 4.1. Meshing

According to the geometric structure of the flow channel and the temperature probe, the 2D tube length L1 = 300 mm and diameter D1 = 20 mm are selected. That is to build a rectangle with a length of 300 mm and a width of 20 mm. A heating wire with length of L2 = 10 mm and diameter of D2 = 4 mm is set in the center of the two-dimensional circular tube, that is, a small rectangle with length of 10 mm and width of 4 mm is set in the center of the large rectangle.

Meshing is very important, which determines the accuracy of the simulation results, but also determines the convergence speed. The denser the meshing, the more accurate the simulation results are. However, too much meshing will bring a lot of calculation. Therefore, according to the actual situation of simulation, appropriate grid should be divided to meet the accuracy of simulation. In this manuscript the boundary of the model is regular, so a quadrilateral mesh is used. In view of this situation, the more detailed the grid division, the more accurate the simulation results in Fluent, but this requires that the computer has enough memory. Therefore, the two sides of the tube in the X-axis direction are divided into 3000 nodes, and the two sides in the Y-axis direction are divided into 200 nodes by GAMBIT. The two sides of the heating wire in the X-axis direction are divided into 100 nodes, and the two sides in the Y-axis direction are divided into 400 nodes. Then the quadrilateral mesh is generated, and the mesh generation is shown in Figure 7.

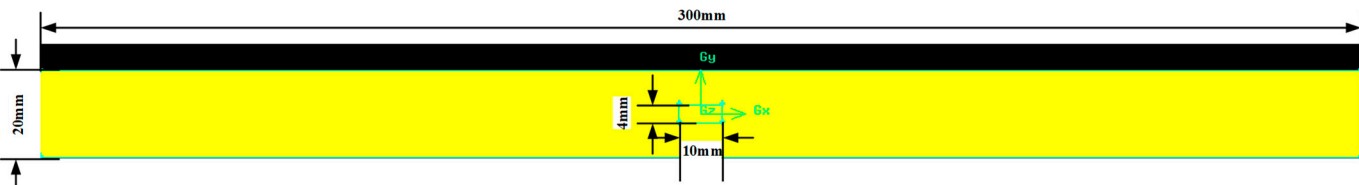

**Figure 7.** Meshing of DTFM.

### 4.2. FLUENT Boundary Condition Setting

The definition of boundary and region is the premise of defining boundary conditions after entering Fluent. After the grid and each calculation area are divided, it is carried out in gambit. The left side of the pipe is the inlet, the inlet type is velocity-inlet, the right side is the outlet, the outlet type is press-outlet and the wall type is wall. The rectangular area drawn by the heating wire is a solid area, and the area type is defined as solid [25–27].

According to the different flow velocity, there are two different flow states for single-phase medium. Laminar and turbulent flow (turbulence). In 1883, Reynold proved through

experiments that the four factors that affect the two flow patterns are combined and called Reynolds number, which is expressed by *Re*:

$$Re = \frac{D\bar{v}\rho}{\mu} = \frac{D\bar{v}}{\gamma} \tag{10}$$

where $D$ is the inner diameter of pipe (m), $\bar{v}$ is the average flow velocity (m/s), $\rho$ is the fluid density (kg/m$^3$), $\gamma$ is the kinematic viscosity (m$^2$/s), $\mu$ is the fluid viscosity (Pa·s).

In addition, the experimental results show that when the value of *Re* is less than 2000, it is laminar flow; when the value of *Re* is greater than 5000, it is turbulent flow; and when the value of *Re* is among 2000 and 50,000, it is transitional flow. $\bar{v}$ is satisfied the following equation.

$$\bar{v} = \frac{4q}{\pi D^2} \tag{11}$$

Reynolds number reflects the contrast between viscous force and inertial force. The larger the Reynolds number is, the more dominant the inertial force is. The smaller the Reynolds number, the higher the viscous resistance and laminar flow.

(1) Laminar flow

When the liquid is at 1 m$^3$/d and the flow flows through a circular pipe with an inner diameter of 20 mm, the cross-section velocity from the inlet direction into the horizontal pipe can be calculated to be 0.0368 m/s, and *Re* = 824, so the flow state in the pipe is laminar.

Assuming the viscosity of water is constant (kinematic viscosity coefficient v = 1.003 × 10$^{-6}$ m$^2$/s), incompressible fluid, smooth round tube, the governing equation of the flow is as follows

(a)　Mass conservation equation:

$$\frac{\partial \rho}{\partial t} + \frac{\partial(\rho u)}{\partial x} + \frac{\partial(\rho v)}{\partial y} + \frac{\partial(\rho w)}{\partial z} = 0 \tag{12}$$

(b)　Momentum conservation equation:

$$\frac{\partial(\rho u)}{\partial t} + \frac{\partial(\rho u u)}{\partial x} + \frac{\partial(\rho u v)}{\partial y} + \frac{\partial(\rho u w)}{\partial z} = \frac{\partial}{\partial x}(\mu \frac{\partial u}{\partial x}) + \frac{\partial}{\partial y}(\mu \frac{\partial u}{\partial y}) + \frac{\partial}{\partial z}(\mu \frac{\partial u}{\partial z}) - \frac{\partial p}{\partial x} \tag{13}$$

$$\frac{\partial(\rho v)}{\partial t} + \frac{\partial(\rho v u)}{\partial x} + \frac{\partial(\rho v v)}{\partial y} + \frac{\partial(\rho v w)}{\partial z} = \frac{\partial}{\partial x}(\mu \frac{\partial v}{\partial x}) + \frac{\partial}{\partial y}(\mu \frac{\partial v}{\partial y}) + \frac{\partial}{\partial z}(\mu \frac{\partial v}{\partial z}) - \frac{\partial p}{\partial y} \tag{14}$$

$$\frac{\partial(\rho w)}{\partial t} + \frac{\partial(\rho w u)}{\partial x} + \frac{\partial(\rho w v)}{\partial y} + \frac{\partial(\rho w w)}{\partial z} = \frac{\partial}{\partial x}(\mu \frac{\partial w}{\partial x}) + \frac{\partial}{\partial y}(\mu \frac{\partial w}{\partial y}) + \frac{\partial}{\partial z}(\mu \frac{\partial w}{\partial z}) - \frac{\partial p}{\partial z} \tag{15}$$

where $u,v,w$ is the component of the velocity vector in x, y and z directions, $\rho$ is the density and $p$ is the pressure on the fluid micro element.

Equation solving: for slender pipe flow, the two-dimensional single precision solver is used in Fluent, and the simple algorithm is used in flow field calculation, which is a kind of pressure correction method.

(2) Turbulent flow

When the liquid is at 3 m$^3$/d and the flow flows through a circular pipe with an inner diameter of 20 mm, the cross-section velocity of the flow into the pipe from the inlet direction is calculated to be 0.1105 m/s and then the Reynolds number is 2473, so the flow in the pipe is greater than 3 m$^3$/d, the flow state is turbulent.

Assuming the viscosity of water is constant (kinematic viscosity coefficient v = 1.003 × 10$^{-6}$ m$^2$/s), incompressible fluid, smooth round tube, the governing equation of the flow is as follows.

(a) Mass conservation equation:

$$\frac{\partial \rho}{\partial t} + \frac{\partial (\rho u)}{\partial x} + \frac{\partial (\rho v)}{\partial y} + \frac{\partial (\rho w)}{\partial z} = 0 \tag{16}$$

(b) Momentum conservation equation:

$$\frac{\partial (\rho u)}{\partial t} + \frac{\partial (\rho uu)}{\partial x} + \frac{\partial (\rho uv)}{\partial y} + \frac{\partial (\rho uw)}{\partial z} = \frac{\partial}{\partial x}(\mu \frac{\partial u}{\partial x}) + \frac{\partial}{\partial y}(\mu \frac{\partial u}{\partial y}) + \frac{\partial}{\partial z}(\mu \frac{\partial u}{\partial z}) + [-\frac{\partial (\rho u'^2)}{\partial x} - \frac{\partial (\rho u'v')}{\partial y} - \frac{\partial (\rho u'w')}{\partial z}] - \frac{\partial p}{\partial x} \tag{17}$$

$$\frac{\partial (\rho v)}{\partial t} + \frac{\partial (\rho vu)}{\partial x} + \frac{\partial (\rho vv)}{\partial y} + \frac{\partial (\rho vw)}{\partial z} = \frac{\partial}{\partial x}(\mu \frac{\partial v}{\partial x}) + \frac{\partial}{\partial y}(\mu \frac{\partial v}{\partial y}) + \frac{\partial}{\partial z}(\mu \frac{\partial v}{\partial z}) + [-\frac{\partial (\rho u'v')}{\partial x} - \frac{\partial (\rho v'^2)}{\partial y} - \frac{\partial (\rho v'w')}{\partial z}] - \frac{\partial p}{\partial y} \tag{18}$$

$$\frac{\partial (\rho w)}{\partial t} + \frac{\partial (\rho wu)}{\partial x} + \frac{\partial (\rho wv)}{\partial y} + \frac{\partial (\rho ww)}{\partial z} = \frac{\partial}{\partial x}(\mu \frac{\partial w}{\partial x}) + \frac{\partial}{\partial y}(\mu \frac{\partial w}{\partial y}) + \frac{\partial}{\partial z}(\mu \frac{\partial w}{\partial z}) + [-\frac{\partial (\rho u'w')}{\partial x} - \frac{\partial (\rho v'w')}{\partial y} - \frac{\partial (\rho w'^2)}{\partial z}] - \frac{\partial p}{\partial z} \tag{19}$$

(c) Turbulent kinetic energy equation:

$$\frac{\partial (\rho k)}{\partial t} + \frac{\partial (\rho ku)}{\partial x} + \frac{\partial (\rho kv)}{\partial y} + \frac{\partial (\rho kw)}{\partial z} = \frac{\partial}{\partial x}[(\mu + \frac{\mu_t}{\sigma_k})\frac{\partial k}{\partial x})] + \frac{\partial}{\partial y}[(\mu + \frac{\mu_t}{\sigma_k})\frac{\partial k}{\partial y})] + \frac{\partial}{\partial z}[(\mu + \frac{\mu_t}{\sigma_k})\frac{\partial k}{\partial z})] + G_k - \rho \varepsilon \tag{20}$$

(d) The dissipation rate equation of turbulent energy:

$$\frac{\partial (\rho \varepsilon)}{\partial t} + \frac{\partial (\rho \varepsilon u)}{\partial x} + \frac{\partial (\rho \varepsilon v)}{\partial y} + \frac{\partial (\rho \varepsilon w)}{\partial z} = \frac{\partial}{\partial x}[(\mu + \frac{\mu_t}{\sigma_k})\frac{\partial \varepsilon}{\partial x})] + \frac{\partial}{\partial y}[(\mu + \frac{\mu_t}{\sigma_k})\frac{\partial \varepsilon}{\partial y})] + \frac{\partial}{\partial z}[(\mu + \frac{\mu_t}{\sigma_k})\frac{\partial \varepsilon}{\partial z})] + \frac{C_{1\varepsilon}\varepsilon}{k}G_k - C_{2\varepsilon}\rho\frac{\varepsilon^2}{k} \tag{21}$$

where $u,v,w$ is the component of the velocity vector in x, y and z directions, $\rho$ is the density and $p$ is the pressure on the fluid micro element.

Combined with the actual situation of this paper, when the fluid flow state is turbulent, the solution equation used is unsteady flow, algorithm, double precision solver and standard $k - \varepsilon$ model.

Before the research, the model assumes are following:

1. The temperature of the fluid in the pipe is constant at 25 °C, and it is a non-Newtonian fluid.
2. The inner wall of the casing is smooth, the temperature is 25 °C and there is no heat exchange with the fluid.
3. The flow pattern of the fluid is calculated according to the calculation formula of Reynolds number. The fluid is single-phase oil and single-phase water, and the physical parameters are shown in Table 1.
4. The structure of the temperature sensor is simplified as a rectangle of 4 mm × 10 mm, and its material is nickel chromium alloy. The physical parameters are shown in Table 1.
5. Since the temperature sensor is set to correct the background temperature, the use of Fluent simulation is mainly to get the temperature changes, so the model of the temperature sensor has not been established.

**Table 1.** Physical parameters of simulation medium.

| Material | Density (kg/m³) | Cp (j/kg·k) | Thermal Conductivity (w/m·k) | Viscosity (kg/m·s) |
|---|---|---|---|---|
| Water | 998.2 | 4182 | 0.6 | 0.001003 |
| Oil | 879.625 | 1857 | 0.127 | 0.0392712 |
| Heating wire | 7200 | 450 | 16.8 | - |

### 4.3. Single-Phase Water Medium

When the heating power of heating wire is 10 W, 20 W and 36 W, respectively, the initial temperature of water is set at 25 °C (298 K), and the temperature rise of water with different flow rate in the vertical upward tube is observed. Figure 7 is the top view of the temperature sensor in Figure 5. Furthermore, Figure 8 shows the temperature distribution of single-phase water when the flow is 1 m³/d, 3 m³/d, 5 m³/d, 10 m³/d, 15 m³/d, 20 m³/d, 25 m³/d, 30 m³/d and the heating power is 10 W. The ordinate represents the Kelvin temperature (units: K). It can be seen that the residual temperature on the heating wire was lower with the increase of flow rate.

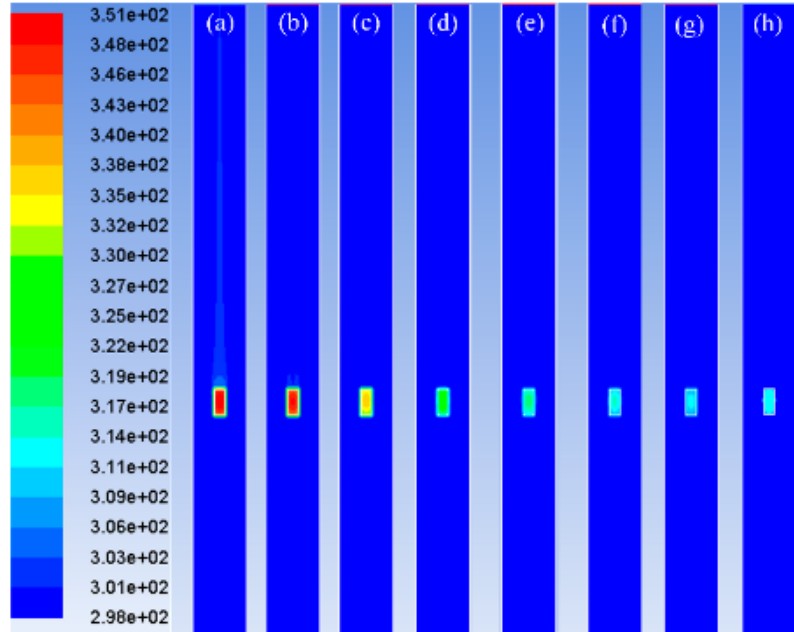

**Figure 8.** Temperature distribution of heating wire at the flow rates as (**a**) 1 m³/d, (**b**) 3 m³/d, (**c**) 5 m³/d, (**d**) 10 m³/d, (**e**) 15 m³/d, (**f**) 20 m³/d, (**g**) 25 m³/d and (**h**) 30 m³/d.

Table 2 shows the temperature value of the measuring point in the numerical simulation of the thermistor, and Figure 8 shows the distribution curve of fluid flow and temperature rise. The abscissa represents the flow (unit: m³/d) which means volume of fluid flowing through each day and the ordinate represents the temperature rise (unit: °C). As shown in the Figure 9, the temperature of the measuring point decreases exponentially with the flow rate of the fluid, and when the flow rate is less than 20 m³/d, it has a good flow resolution, and when the flow rate is greater than 20 m³/d, the resolution decreases.

**Table 2.** Temperature distribution data of heating wire at heating power is 10 W.

| Flow (m³/d) | 1 | 3 | 5 | 10 | 15 | 20 | 25 | 30 |
|---|---|---|---|---|---|---|---|---|
| Initial fluid temperature (°C) | 25 | 25 | 25 | 25 | 25 | 25 | 25 | 25 |
| Measuring point temperature (°C) | 30.18 | 28.88 | 27.74 | 26.7 | 26.33 | 26.13 | 26.01 | 25.93 |
| Temperature Rise (°C) | 5.18 | 3.88 | 2.74 | 1.7 | 1.33 | 1.13 | 1.01 | 0.93 |

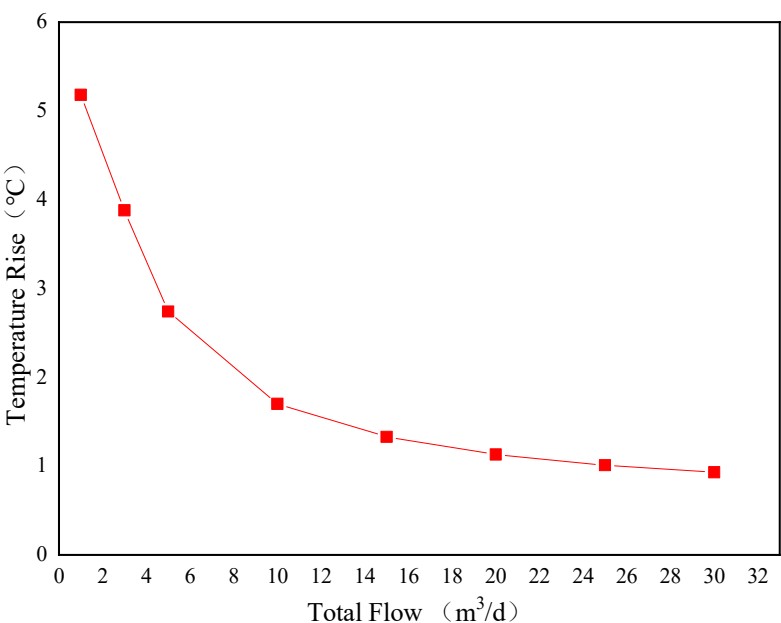

**Figure 9.** Curve diagram of temperature rise and flow rate variation at heating power 10 W.

Figure 10 shows when the heating power is 10 W, 20 W, 36 W, the relationship between temperature rise and flow rate. The relationship between temperature rise and flow rate is exponential decline, the trend of curve decreases with the increase of flow rate. The larger the heating power is, the larger the temperature rise is and the higher the resolution is. When the flow rate is higher than 20 m$^3$/d, the curve tends to be flat and the flow resolution decreases. When the flow rate is less than 20 m$^3$/d, the curve changes obviously and has a good flow resolution. The higher the heating power, the higher the resolution.

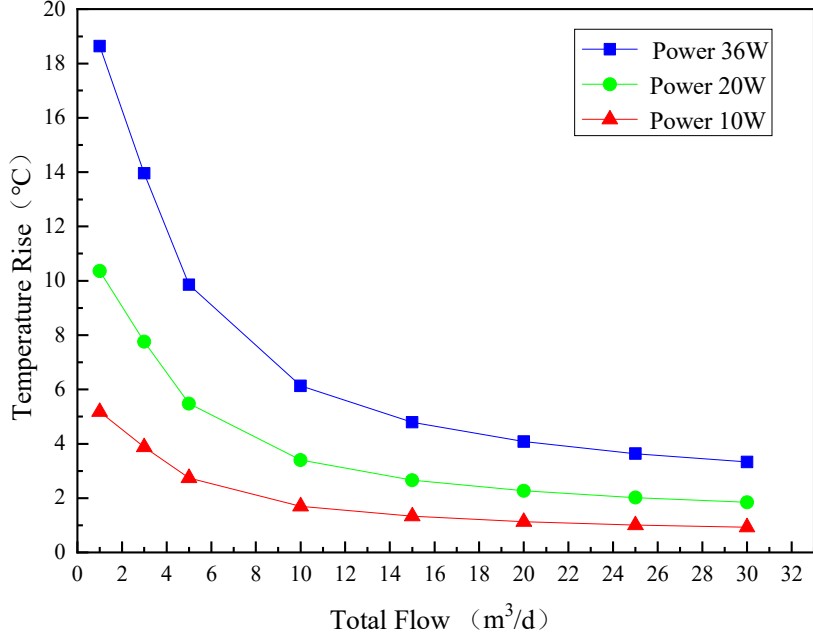

**Figure 10.** Curve diagram of temperature rise and water flow rate variation at different heating power.

### 4.4. Single-Phase Oil Medium

Changing the medium in the pipe to oil, and the heating power of heating wire is 10 W, 20 W and 36 W, respectively. The initial temperature of oil in the pipe is set to 25 °C (298 K), and the other settings are the same as when the medium is single-phase water. As shown in Figure 11, the law curve of temperature distribution trend and flow rate when the medium is oil. The overall trend decreases with the increase of flow rate, the temperature rise decreases and the degree of reduction is different. The higher the heating power, the greater the temperature rise; the smaller the heating power, the smoother the curve of temperature change. It can also be seen that the curve is almost horizontal when the flow rate is more than 20 m$^3$/d, and the resolution is low at this time. The curve changes significantly when the flow rate is less than 20 m$^3$/d, and there is a good flow resolution. It can be seen that this device is more accurate for measuring low flow.

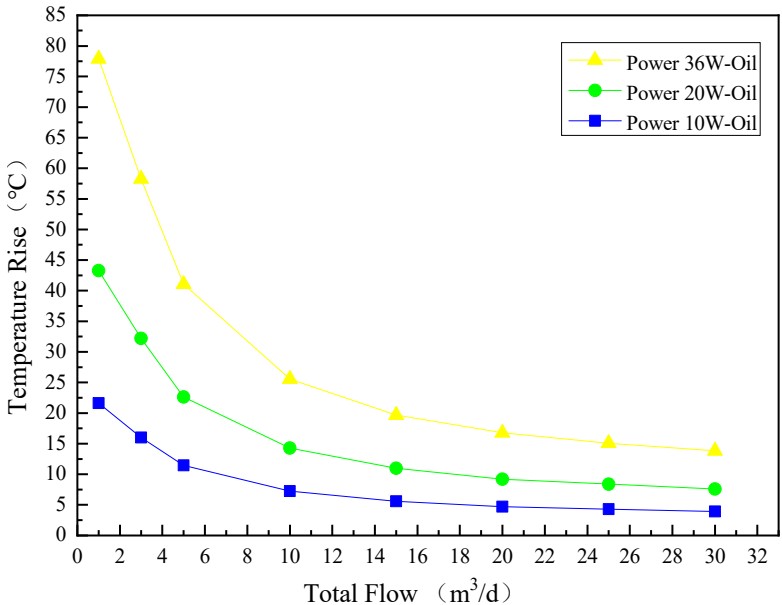

**Figure 11.** Curve diagram of temperature rise and oil flow rate variation at different heating power.

### 4.5. Result Analysis at Different Medium

The data obtained when the medium is single-phase water and single-phase oil are comprehensively analyzed, as shown in Figure 12. From top to bottom, the corresponding temperature rise of 36 W oil, 20 W oil, 10 W oil, 36 W water, 20 W water and 10 W water are shown. It can be seen from the simulation results that the overall trend of temperature rise of heating wire decreases with the increase of flow rate. The overall temperature rise of oil is higher than that of water. The reason is that the specific heat capacity of oil is smaller than that of water, it absorbs the same heat and the temperature changes greatly. The simulation results prove this point. At the same power, the difference between oil and water in Y-axis is different, and its resolution is obvious. Whether it is water or oil, the larger the heating power of the heating wire, the steeper the curve and the higher the resolution; the smaller the heating rate of the heating wire, the smoother the curve. When the flow rate is less than 20 m$^3$/d, the curve changes obviously and has a good resolution. When the flow rate is more than 20 m$^3$/d, the curve is almost parallel to the X-axis and the flow resolution is low. Therefore, the simulation results show that when using thermal mass flowmeter to measure oil-water single-phase flow, it is suitable to measure low flow.

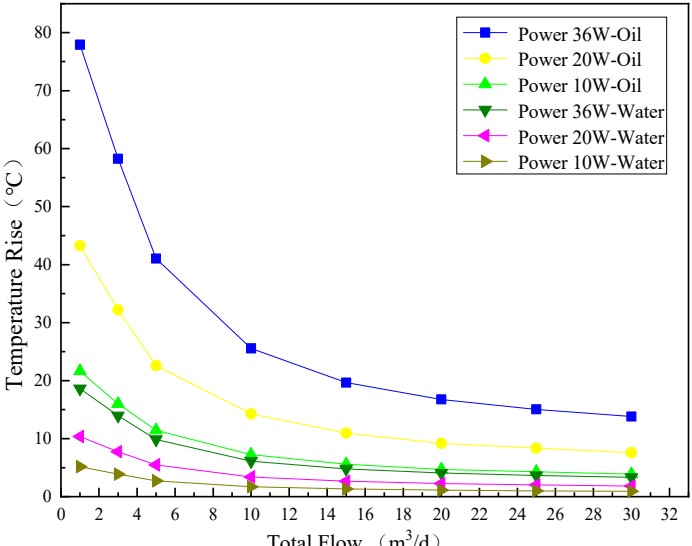

**Figure 12.** Temperature rise curve in the medium of single-phase oil and single-phase water.

### 4.6. Study on Steady State Time of Heating Wire Temperature Rise

4.6.1. Simulation Results and Data Analysis in Single-Phase Water

Set the physical parameters of the input fluid as water in Fluent, and the power is 36 W, and the input flow rates are 1 m³/d, 3 m³/d, 5 m³/d, 10 m³/d, 15 m³/d, 20 m³/d, 25 m³/d and 30 m³/d for simulation. When the heating wire is heated to 5 s, 10 s, 20 s, 40 s, 60 s, 80 s and 90 s, the temperature rise value were shown in Figure 13. The abscissa is time (units: s), and the ordinate is the temperature rise (units: °C). It can be seen from the curve in the figure that when the flow rate is constant, the temperature change with the times. According the curves data, when the flow rate is 1 m³/d, the temperature rise is maximum. Furthermore, the temperature rise description is the same at different flow rates. The rising trend is that the temperature rise is faster in the heating time of 40 s, and slower when the heating time is greater than 40 s. After 40 s, the temperature rise is basically stable, the change of temperature rise was not obvious. It is obvious from the 1 m³/d flow point that the temperature rise changes the most in the period of 1–10 s. When the flow rate is increased, the sensitivity of the DTFM decreases and the temperature change decreases gradually.

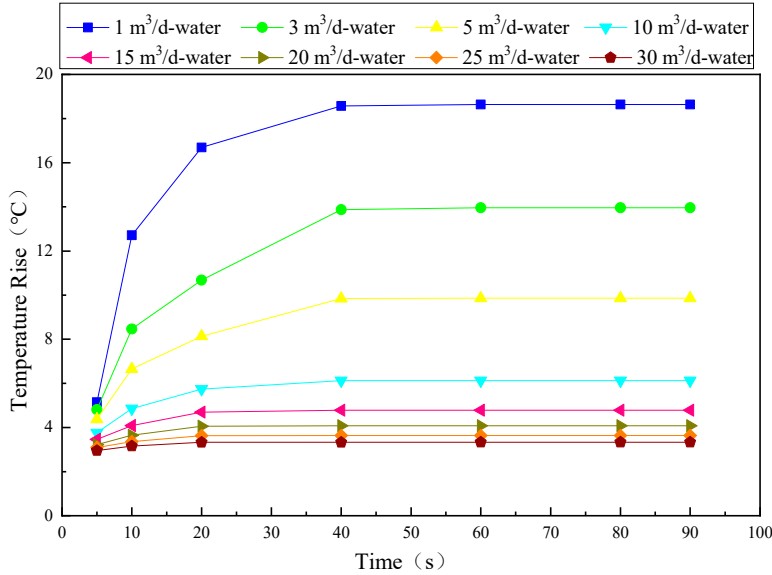

**Figure 13.** Curve of temperature rise with time at different flow rates in water.

Through the simulation research on the heating time of the heating wire in the pipe, the heating time of the heating wire can be set as 40 s, because when the power is fixed and the heating time is longer than 40 s, the temperature rise is basically unchanged. It has a good guiding significance for the simulation wellbore experiment.

### 4.6.2. Simulation Results and Data Analysis in Single-Phase Oil

Setting the input phase as oil, the power as 36 W and the inlet flow rates as $1\ m^3/d$, $3\ m^3/d$, $5\ m^3/d$, $10\ m^3/d$, $15\ m^3/d$, $20\ m^3/d$, $25\ m^3/d$ and $30\ m^3/d$, respectively, for simulation. When the heating wire is heated to 5 s, 10 s, 20 s, 40 s, 60 s, 80 s and 90 s, monitor the temperature rise and draw the temperature rise calculated by simulation as shown in Figure 13. It can be seen from figure that it is similar to the curve when the medium is water. In the simulation, the medium in the pipe is changed to oil, and the relevant parameters such as density and viscosity are also included. The temperature rise is different at different times under different flow rates, but the rising trend of temperature is basically the same as that in water. The value of temperature rise is different, and the value in oil is obviously greater than that in water. When the heating time is within 40 s, the temperature rise is faster than 40 s and it is slower than 40 s. After 40 s, the temperature rise is basically in a stable state, but there are also small changes.

### 4.6.3. Comparative Analysis of Simulation Results in Water and Oil

According to the above analysis of temperature rise with time when the medium is water and oil, the temperature rise trend of water and oil is basically the same, but the temperature rise of oil is much higher than that of water. In Figure 14, when the heating time is from 0 s to 10 s, the temperature rise is the fastest. The temperature rise of 10–20 s is also fast, the temperature rise of 20–40 s changes slowly and the temperature rise after 40 s changes little. After 60 s of simulation, the temperature rise of water and oil basically remains unchanged. This simulation has a great guiding significance to the simulation well Dynamic experimental, and avoids the shortcomings of not knowing when the temperature is stable and waiting time is too long.

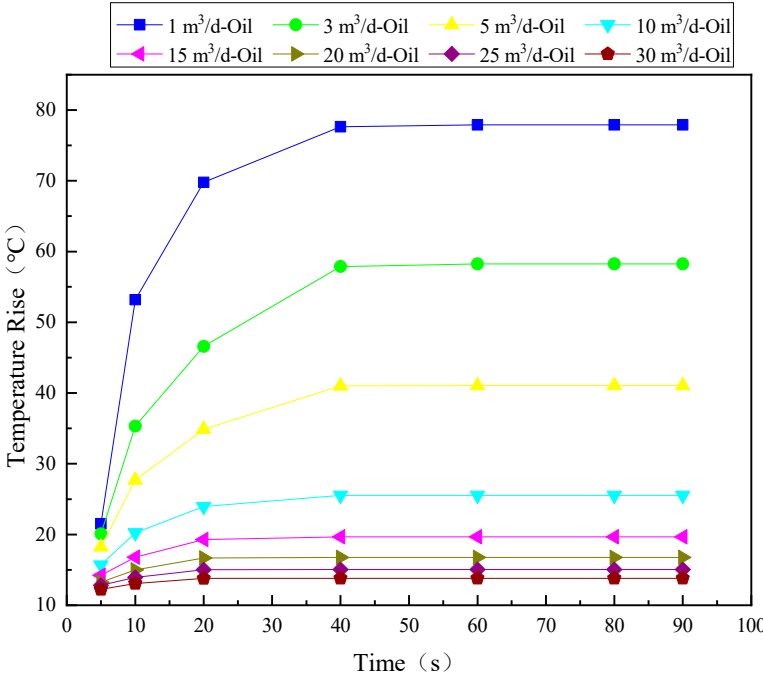

**Figure 14.** Curve of temperature rise with time at different flow rates in oil.

### 4.6.4. Comparative Analysis of Simulation Results in Water and Oil

According to the above analysis of temperature rise with time when the medium is water and oil, the temperature rise trend of water and oil is basically the same, but the temperature rise of oil is much higher than that of water. When the heating time is from 0 s to 10 s, the temperature rise is the fastest. The temperature rise of 10–20 s is also fast, the temperature rise of 20–40 s changes slowly and the temperature rise after 40 s changes little. After 60 s of simulation, the temperature rise of water and oil basically remains unchanged. This simulation has a great guiding significance to the simulation well Dynamic experimental, and avoids the shortcomings of not knowing when the temperature is stable and waiting time is too long.

## 5. Dynamic Experimental

### 5.1. Multiphase Flow Simulation Experimental Facility

In this paper, the Dynamic experimental of oil-gas-water multiphase flow were carried out on the experimental facility of oil-water two-phase flow in Daqing Oilfield, which consists of transparent simulated wellbore with length of 8 m, inner diameter of 125 mm, overhead tank, oil tank, water tank, water pump, oil pump, oil-water separation tank, air compressor, gas purification device, gas stable pressure tank and the console, as shown in Figure 15. The blue simulated wellbore is the experimental wellbore, the pink wellbore is the recycle wellbore. Finally, water and oil flow into the recovery wellbore through the experimental wellbore, forming a circulation system.

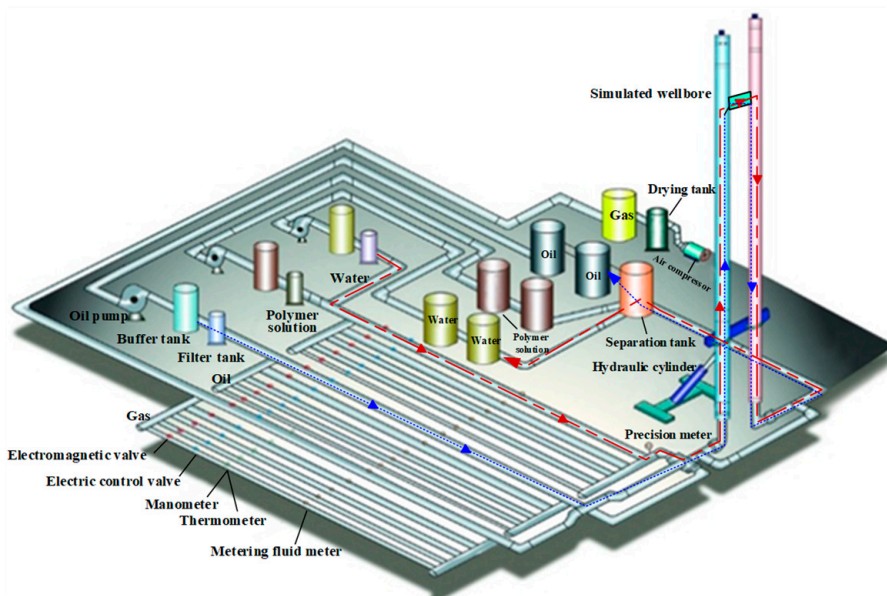

**Figure 15.** Working sketch of the multiphase flow simulation experimental facility.

In the Figure 15, oil tank and water tank stored the diesel and water used in the experiment, oil pump and water pump inhaled liquid from the oil tank and water tank, and pumped into the respective overhead tank on derrick. Oil-water two-phase flow under the control of their respective control, through the manual ball valve, DTFM and automatic ball valve together into the vertical upward simulated wellbore. The red dotted line is the water direction of water flow and the blue dotted line is the oil direction of oil flow. Figure 16 shows the DTFM is in the simulated wellbore.

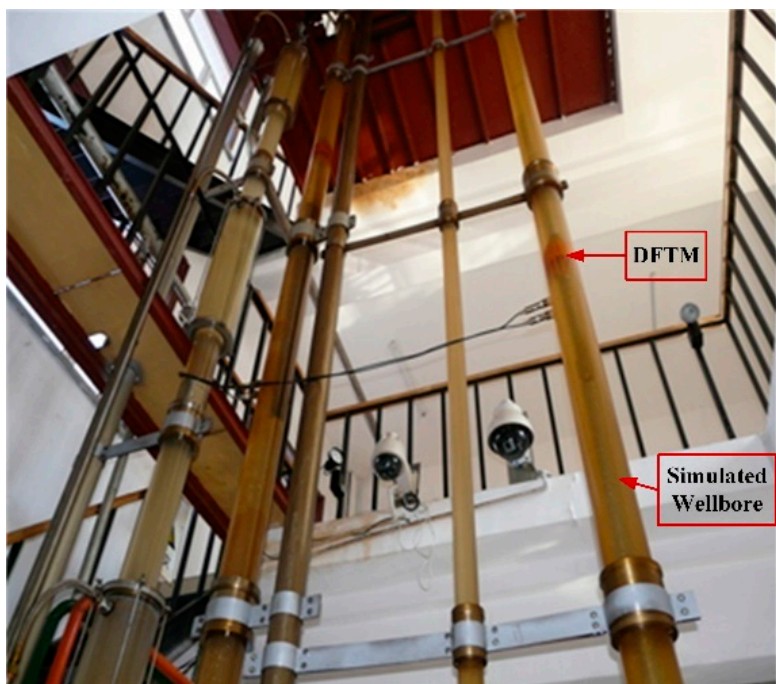

**Figure 16.** DTFM in the simulated wellbore.

*5.2. Dynamic Experimental of DTFM on Multiphase Flow Facility*

The experimental research was carried out on the multiphase flow simulation experimental facility of Daqing Oilfield testing technology service company by using the principle experimental prototype of DTFM. The inner diameter of the vertical simulation wellbore is 125 mm. In the experiment, the collecting umbrella is opened to close the annular space between the prototype and the well bore, so that all fluid flows through the measurement channel of the prototype and the fluid velocity is increased.

In order to understand the response law of the probe with temperature correction in the fluid flow state, the Dynamic experimental was carried out in the multiphase flow simulation experimental facility. The experimental medium was oil and water, and the flow rates were 1 m$^3$/d, 3 m$^3$/d, 5 m$^3$/d, 7 m$^3$/d, 10 m$^3$/d, 20 m$^3$/d, 25 m$^3$/d, 30 m$^3$/d, 35 m$^3$/d and 40 m$^3$/d. Experiments were carried out in single-phase water and single-phase oil, respectively, to measure the variation of voltage with flow under different flow rates to verify the repeatability and stability of the measurement device. In the Dynamic experimental, the liquid flow is adjusted one by one from large to small, the first flow is 40 m$^3$/d, and the last is 1 m$^3$/d. Supply 12V 3A (36W) direct current (DC) to the resistance heating wire.

5.2.1. The Experiment of Single-Phase Water and Single-Phase Medium

The output voltage is obtained through experimental research, and the relationship curve between voltage and temperature rise is obtained according to Equations (8) and (9), as shown in the Figure 17. The curve of temperature rise versus flow rate measured two times is given. Water 1 is the first test and water 2 is the repeated test. Oil 1 is the first test and the oil 2 is the repeated test. For the water curves, these can be seen from the figure that the repeatability of the two measurements are good, and the data has a certain fluctuation. When the flow rate is low, the resolution is very good. When the flow rate is below 10 m$^3$/d, the voltage changes obviously. After 20 m$^3$/d, the deviation is large. The overall trend of the data curve is good. For the oil curves, the flow resolution when measuring the oil is better than that of the water. The reason is that the heat capacity of oil is small and the heat conduction coefficient is small; the flow rate is in the range of 1 m$^3$/d ~ 30 m$^3$/d. There is a certain error in the repeatability of the two measurements. On the

one hand, when the medium is oil, there is contaminant oil on the experimental instrument, which affects the measurement results. On the other hand, because the measurement is carried out on the simulated well, the experimental environment of the simulated well is closer to the real field environment, so there is a certain error, this is within the tolerance of the error.

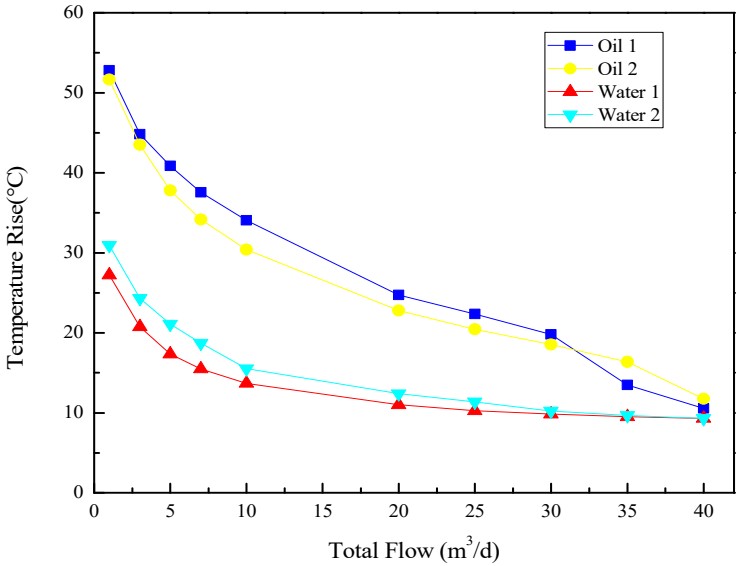

**Figure 17.** Curve of temperature rise changing with flow rate in water and oil.

5.2.2. Comparison of Dynamic Experiment and Simulation Study

As shown in the Figure 18, the temperature rise of the experimental data is higher than that of the simulation data when the medium is water, but the curve distribution is the same. The difference between the simulation data and the experimental data is that there is a correction factor in the actual measuring instrument. Therefore, it can be known that the simulation data has guiding significance to guide the experimental research.

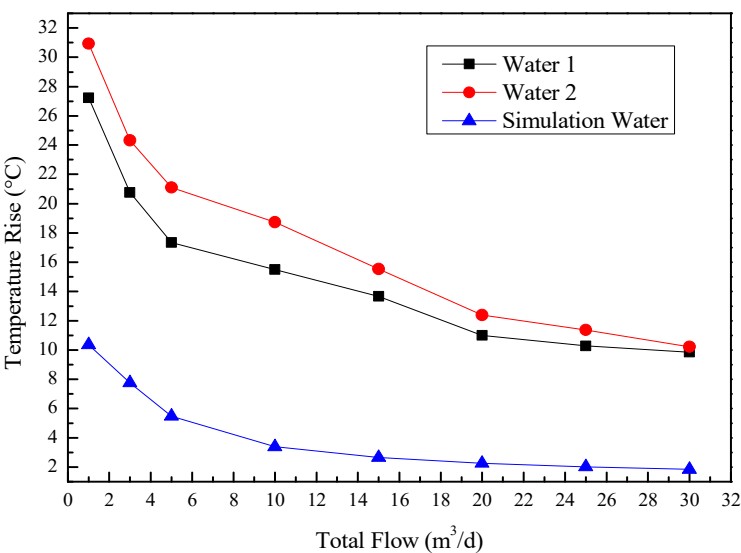

**Figure 18.** Comparison of dynamic experimental data and simulation data at water.

Figure 19 represents that the data comparison diagram when the medium is oil. It can be seen that there are some differences between the simulation data and the experimental data. The main reason is that the temperature of the oil used in the dynamic experiment is easy to be disturbed by the outside world, so the temperature cannot be kept constant. Especially when the flow rate is small, the measuring instruments are more sensitive to temperature, so there are differences, but the distribution trend is consistent with the theoretical analysis. Thus, it can be concluded that the simulation analysis has a certain guiding significance for the experimental research.

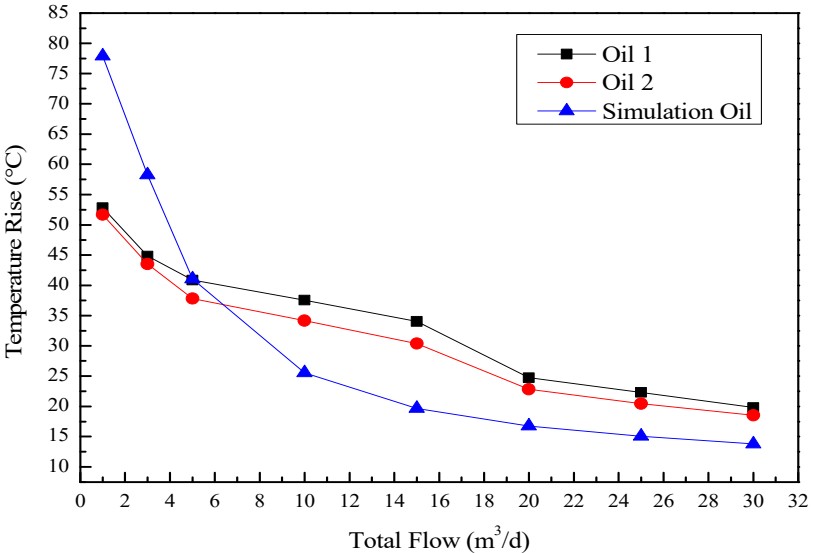

**Figure 19.** Comparison of dynamic experimental data and simulation data at oil.

## 6. Conclusions

In this paper, a new method DTFM is proposed for measure the flow rate of liquid phase underground. Studied theoretically and experimentally of DTFM at the relationship between heating power and flow rate. The response law between temperature and flow rate is obtained. Finally, the dynamic experimental of DTFM used in downhole flow measurement is carried out. The conclusion is as follows:

1. The temperature field inside the measuring device is simulated by Fluent. The temperature rise with time is obtained. With the increase of flow rate, the temperature rise on the DTFM changes less. The larger the heating power is, the more obvious the temperature field changes. However, when the flow rate is less than 20 m$^3$/d, the resolution of the sensor is higher. When the flow rate is more than 20 m$^3$/d, the resolution decreases. The accurate simulation results provide a theoretical basis for dynamic experimental research.

2. The relationship between the heating time and the heating power is obtained through the simulation experiment. Whether the medium is water or oil, the heating time of the heating wire can be considered as reaching the steady state after 40s, which provides the theoretical results of the heating time for the dynamic experimental.

3. Dynamic experimental is carried out on a multiphase flow simulation experimental facility. The results show that when the medium is single-phase water and single-phase oil, the DTFM has good response characteristics when the flow rate is less than 20 m$^3$/d and has good flow resolution. This is the same as the result of numerical simulation, and also shows that the DTFM is suitable for low flow measurement. It provides a theoretical basis for practical application.

4. The research of theoretical and dynamic experimental proves that it is feasible to use the thermal flowmeter to measure the flow rate of underground liquid. The research of DTFM provides a new idea with the characteristics of high reliability, no movable

parts, high accuracy of low flow measurement and small disturbance to fluid for downhole flow measurement, which has a strong application value.

**Author Contributions:** Conceptualization, Y.Y. and L.H.; methodology, X.L.; software, L.Y. and W.L.; validation, Y.Y. and L.H.; formal analysis, W.L.; data curation, X.L. and W.L.; writing—original draft preparation, Y.Y. and L.H.; writing—review and editing, Y.Y. and L.H.; visualization, Y.Y., Z.J. and L.H.; project administration, L.Y., Z.J. and X.L.; funding acquisition, L.H and X.L. All authors have read and agreed to the published version of the manuscript.

**Funding:** This research was funded by" Northeast Petroleum University Youth National Fund, grant number 2017QNJL-07"," the National Natural Science Foundation of China, grant number 51774092"," National Natural Science Foundation of Heilongjiang Province, grant number LH2020E012"," the China Postdoctoral Science Foundation, grant number 2016M601399"and "the CNPC project, grant number 2019d-3807".

**Data Availability Statement:** The data presented in this study are available on request from the corresponding author.

**Acknowledgments:** The authors would like to thank Zhao tiezhu (Logging and Testing Services Company, Daqing Oilfield Limited Company) for his help about experiment.

**Conflicts of Interest:** The authors declare no conflict of interest.

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
