# Peer review of "Flow Rate Measurement of Production Profile Logging Using Thermal Method"

_water, doi:10.3390/w13111544_

Round 1

Reviewer 1 Report

In the introduction the Authors should concentrate on typical multiphase flow measurement methods. It is obvious, that turbine or electromagnetic meters are inappropriate for such measurements.

Because of better properties practically all the thermal mass flowmeters are  manufactured on the basis of constant temperature difference principle. Application of the constant current principle should be justified.

Many terminology errors, examples: Pt 1000 is not thermistor. This terms are mistakenly used interchangeably over the paper. Temperature unit is Kelvin not Calvin. To be precise:  collecting umbrella can not increase the flow rate.

There is no information about crucial CFD simulation parameters, such as mesh discretization scheme, grid independence test, boundary and initial conditions, solver and turbulence model used, 3D or 2D.

Fig. 6 is illegible. Should be zoomed at the temperature sensor to be more clear for the reader. Moreover, fig. 6 does not correspond with fig. 5. I would expect the numerical model to resemble (have the same geometry) the temperature sensors system presented in fig. 5. I am not able to understand how represents the rectangle in fig. 6 the sensor presented in fig. 5.

Fig. 7-9 are redundant, fig. 10 contains the complete information. The same for fig. 17 and 18, they are redundant, fig. 19 contains the complete information.

Experimental part: Fig. 15 is not clear or not complete, where the fluids from the simulated wellbore are evacuated ?

  1. 5.2.1 - difference between 2 measurement is not good measure of repeatability. A series of more numerous measurements measurement should be carried out.

A great disadvantage is that experiments results are presented in terms of voltage and CFD simulations in terms of temperature rise. It makes it impossible to compare results obtained by both methods. It could be both interesting and enable the validation of CFD simulations. The experimental results should be recalculated from voltage to temperature difference, the electrical scheme of the temperature measurement circuit (bridge?) could also be presented.

Some additional value could contribute calculations from “analog” formulae 1  - 7 and comparison of results with other methods. By the way – these formulae are not the Authors authorship, so the source must be given in references.

The paper describes new approach to the measurement of liquid - liquid two phase flow. The CFD and experimental results concern however only one phase flow, water or oil, there are no information how the method would work with the mixture and how to distinguish the output signal  change due to  velocity changes or composition of mixture change. It is difficult to assess if the method has the potential to measure measure the individual components flow rate in the mixture.

Author Response

Dear reviewer,

We would  like to thank you for their efforts in reviewing and improving our manuscript. Our replies  to your the comments and questions are in the attachment. We hope that our revisions can satisfy  you.

If you have any queries, please don’t hesitate to contact me at the address below.
Thank you and best regards.
Yours sincerely,
Lianfu Han;
Corresponding author:
Name: Lianfu Han
E-mail: [email protected]

Reviewer 2 Report

The article presented for review concerns the measurement of oil-water flow in oil-bearing wells. The subject of the article is up-to-date, the presented design of the flow meter is innovative. There are several inaccuracies in the article that should be cleared up.

  1. In the descriptive part of paper, the described method is analogous to the Hot Wire flow meters. Why did the authors not refer to this method. It should be clarified what are the differences between the two methods., the authors did not mention the impedance tomography method as an alternative solution. It would be worth referring to.
  2. The authors use the unit m3 / d, please explain what d means. Usually we use m3 / h, m3 / min e.t.c.
  3. Who is it about: the Calvin type temperature, is it not about Kelvin?
  4. The obtained results of simulations and tests are quite obvious. They suggest that only low flow measurements are possible it is truth .

Author Response

(The authors gave the same response as above.)

Round 2

Reviewer 1 Report

Some my remarks were taken into accounts. But the main doubts remained.

The development of the subject of CFD simulations showed the poor meshing, 2D technique for 3D object, uniform meshing is not adequate to the problem, lack of boundary layer mesh and mesh adaptation in the sensors region can significantly distort the results or even give erroneous results. Probably for this reason the authors did not “zoomed” at the sensors region.

I can also not accept the Authors denied to recalculate voltage to temperature difference. It is an elementary task, to calculate RTD resistance from voltage (Ohm law) and then calculate the temperatures indicated by both sensors from ISO/IEC 17025. It would be valuable contribution to CFD validation.

The title, first concerning two phase mixture measurement and not adequate to the article content has been changed, but the Authors should at least underline, that this is the very preliminary stage and present some ideas, how to how to distinguish between output signal  change due to  velocity changes and composition of mixture change. There are some possibilities, omitted by the authors in the introduction (microwaves, MEGRA, DEGRA, sonic arrays etc.)

The text in lines 253 – 256 about Re definition is probably too elementary to “Water” readers.

Perhaps after most professional CFD analysis and consistent comparison with experiments and CFD the paper could be published in “Water”, but first some experiments with real two phase flow mixtures would make the research would make this paper of excellent value, especially if the Authors can use the unique stand described in the paper.

Author Response

 Dear reviewer

Thank you very much for your letter and the comments from the referees about our paper submitted to flow rate measurement of production profile logging using thermal method. We have revised the manuscript according to your kind advises and referee’s detailed suggestions. Enclosed please find the responses to the referees. We sincerely hope this manuscript will be finally acceptable to be published on ‘Water’。

If you have any queries, please don’t hesitate to contact me at the address below.
Thank you and best regards.
Yours sincerely,
Lianfu Han;
Corresponding author:
Name: Lianfu Han
E-mail: [email protected]

Round 3

Reviewer 1 Report

Some comparison between CFD and experiments in the conclusions would be of interest.

This manuscript is a resubmission of an earlier submission. The following is a list of the peer review reports and author responses from that submission.